# Enhancing Functionalization of Health Care Textiles with Gold Nanoparticle-Loaded Hydroxyapatite Composites

**DOI:** 10.3390/nano13111752

**Published:** 2023-05-27

**Authors:** Bárbara Vieira, Jorge Padrão, Cátia Alves, Carla Joana Silva, Helena Vilaça, Andrea Zille

**Affiliations:** 1CITEVE, Technological Centre for the Textile & Clothing Industry, 4760-034 Vila Nova de Famalicão, Portugal; 2Centre for Textile Science and Technology, University of Minho, 4800-058 Guimarães, Portugal

**Keywords:** functionalization, knitted textiles, antimicrobial, nanoparticles, washing fastness

## Abstract

Hospitals and nursing home wards are areas prone to the propagation of infections and are of particular concern regarding the spreading of dangerous viruses and multidrug-resistant bacteria (MDRB). MDRB infections comprise approximately 20% of cases in hospitals and nursing homes. Healthcare textiles, such as blankets, are ubiquitous in hospitals and nursing home wards and may be easily shared between patients/users without an adequate pre-cleaning process. Therefore, functionalizing these textiles with antimicrobial properties may considerably reduce the microbial load and prevent the propagation of infections, including MDRB. Blankets are mainly comprised of knitted cotton (CO), polyester (PES), and cotton-polyester (CO–PES). These fabrics were functionalized with novel gold-hydroxyapatite nanoparticles (AuNPs-HAp) that possess antimicrobial properties, due to the presence of the AuNPs’ amine and carboxyl groups, and low propensity to display toxicity. For optimal functionalization of the knitted fabrics, two pre-treatments, four different surfactants, and two incorporation processes were evaluated. Furthermore, exhaustion parameters (time and temperature) were subjected to a design of experiments (DoE) optimization. The concentration of AuNPs-HAp in the fabrics and their washing fastness were critical factors assessed through color difference (ΔE). The best performing knitted fabric was half bleached CO, functionalized using a surfactant combination of Imerol^®^ Jet-B (surfactant A) and Luprintol^®^ Emulsifier PE New (surfactant D) through exhaustion at 70 °C for 10 min. This knitted CO displayed antibacterial properties even after 20 washing cycles, showing its potential to be used in comfort textiles within healthcare environments.

## 1. Introduction

Most of the population of hospitals and nursing homes may be classified as vulnerable to microbial infections. Multidrug-resistant bacteria (MDRB) are among the major concerns of worldwide healthcare units since their growing prevalence represents a serious threat to the Sustainable Development Goals set by the World Health Organization (WHO) [1,2,3]. MDRB infections represent between 20 and 25% of cases in hospitals and approximately 17% in nursing homes [4,5]. In addition, current viral pandemics and outbreaks represent a high concern in healthcare units, as some species can maintain viability for up to 60–140 days on soft surfaces subjected to repeated contact [6]. Therefore, textile materials may represent an important vehicle for the propagation of microorganisms, due to their high contact surface and capacity to retain moisture, creating a perfect environment for the maintenance and spreading of virus and bacteria [3,7,8]. This is particularly relevant in common comfort healthcare textiles ubiquitously present in hospitals and nursing centers, such as blankets. These fabrics can be easily shared without proper hygienization, increasing the prevalence of MDRB and viruses. Although most medical textiles are non-woven (>60%), comfort textiles for healthcare applications, such as blankets, are mainly made of knitted fabric or woven fabric [9]. Knitted fabrics are made from connected loops of yarn and are suitable for comfort textiles due to their flexible, loose structure and thermal properties [9,10]. In contrast to non-woven fabrics, that are cheaper and usually disposable, woven and knitted fabrics have a longer lifetime in the medical care field and are, therefore, more sustainable options [11].

Therefore, the development of antimicrobial textiles is highly relevant for hospitals, nursing homes, and other microbial-prone environments as a strategy to reduce the environmental load of viable microorganisms. Nanotechnology has been highly studied in textile functional finishing due to its diverse advantages, such as lower consumption of active substances, easier introduction into the fabrics as nanoparticles (NPs), high surface area, and reactivity [12]. Typically, metallic microparticles and NPs, such as silver, copper, gold, zinc, and titanium NPs, have been applied in textiles [12,13,14]. Currently, silver NPs (AgNPs) are the most extensively studied and used for the production of antimicrobial textiles, including personal protective equipment, home textiles, and biomedical devices [15,16]. These NPs have robust antimicrobial activity, even at a low concentration. The silver ions can compromise cell wall integrity and generate reactive oxygen species (ROS) that effectively compromise genomic and proteomic activity, promoting cell death [16,17,18,19,20]. However, there is important evidence that they can be toxic to humans and cause environmental issues due to their low fastness to washing and other mechanical actions, such as rubbing [21,22,23,24,25]. Thus, the superior functionalization of textiles that provide enhanced washing fastness to innovative antimicrobial NPs may represent a viable strategy to reduce infections in hospitals and nursing homes [26].

The antimicrobial agent used to functionalize knitted fabrics is nanocomposites patented by the Jožef Stefan Institute (JSI) [27]. The innovative composites comprise gold NPs functionalized with amine and carboxyl groups immobilized in nanohydroxyapatite (AuNPs-HAp). These are less toxic to human cells than AuNPs alone and possess higher chemical stability, and nano-HAp aids NPs’ dispersion [28] and improves the antimicrobial properties of the composite [29,30]. The role of HAp is to promote a homogeneous dispersion preventing the AuNPs’ aggregation, thus providing a uniform functionalization of the textiles. Furthermore, the use of HAp composite is an adequate immobilization substrate for AuNPs, as it displays enhanced biocompatibility.

This study envisaged the optimization of the anchoring of AuNPs-HAp into knitted textiles to provide antibacterial properties with enhanced washing fastness to comfort health care textiles. Therefore, two pre-treatments were tested: atmospheric dielectric barrier discharge (DBD) plasma atmosphere (DBD-plasma), and half-bleaching. For the dispersion of AuNPs-HAp, four surfactants were evaluated, and two incorporation procedures: exhaustion and impregnation (padding). A design of experiments (DoE) was established to optimize functionalization parameters, namely time and temperature.

## 2. Materials and Methods

### 2.1. Materials

The textile knitted fabrics were kindly provided by INNOWEAR-TEX Kft (Hódmezővásárhely, Hungary): i. 100% cotton (CO), raw, polo fabric in single jersey with 180 g/m^2^; ii. 50% CO/50% polyester (PES), water green, polo fabric in single jersey with 165 g/m^2^; iii. 100% PES, white, polo fabric in double needle bird eye with 200 g/m^2^ (Figure 1).

The AuNPs-HAp were kindly provided by the Jožef Stefan Institute (Ljubljana, Slovenia) and produced as described in [26], with arginine (arg) as the organic molecule, and functionalized with amine and carboxyl groups. The received AuNPs-HAp were dispersed in distilled water and sonicated (37 kHz, 80 W) for 1 h to homogenize the distribution of the AuNPs-HAp, which exhibited a concentration of 0.4 mg/mL. The tested surfactants were Imerol^®^ Jet-B (surfactant A) and Leonil^®^ EHC (surfactant B) from Archroma (Pratteln, Switzerland), Emulsifier WN from Tanatex Chemicals (Ede, The Netherlands) (surfactant C), and Luprintol^®^ Emulsifier PE New from Archroma (surfactant D). Other reactants were ECE non-phosphate reference detergent from SDC Enterprises Ltd (Thongsbridge, UK); Diadavin UN and Tannex GEO from Tanatex Chemicals; sodium hydroxide (50% (*w*/*v*)), hydrogen peroxide 200 vol. 50% (*v*/*v*), and sodium carbonate from João Manuel Lopes de Barros, Lda (Maia, Portugal); acetic acid from Thermo Fisher Scientific (Waltham, MA, USA); and glutaraldehyde 25% (*v*/*v*), grade II, and citric acid from Sigma-Aldrich (Darmstadt, Germany).

### 2.2. Pre-Treatments

The pre-treatments applied to CO were half-bleaching and DBD plasma. CO–PES and PES pre-treatment was DBD plasma. 

Half-bleaching: The process was carried out in a jet (Mathis Labor Jumbo Jet JFO; Oberhasli, Switzerland), where the fabric sample was put with the bleaching mixture (0.5 g/L Diadavin UN, 0.5 g/L Tannex GEO, 2 mL/L caustic soda at 50% (*w*/*v*), and 3 mL/L hydrogen peroxide 200 vol. 50% (*v*/*v*)) at 98 °C for 30 min. Subsequently, the hydrogen peroxide was neutralized with a bath containing 1 g/L of Baylas EFR at 50 °C for 10 min, followed by a rinse bath to drop the pH to ≈6, containing 1 g/L acetic acid at 50 °C for 10 min.

Plasma: The fabrics were pre-treated using a DBD atmospheric pressure plasma equipment (semi-industrial equipment; Softal Electronics GmbH, Hamburg, Germany). In this application, it was not used as a reactive gas. The textile sample passed between two metallic electrodes, at a power of around 1 kW and a speed of 4 m/min [31,32].

### 2.3. Textiles Functionalization

Padding: AuNPs-HAp dispersions were used in a liquor-to-material ratio of 20:1, and the sample was immersed in the suspension for 2 min. The process was carried out in a Mathis Padder HVF (Oberhasli, Switzerland) and the pressure between rollers was adapted for a liquor uptake of 100%. 

Exhaustion: To the prepared AuNPs-HAp suspensions, additives (if any) were added as described in Table 1. The pH was adjusted to 9 and other variables were analyzed (bath ratio, time, temperature, and additives). The exhaustion was performed in a Mathis Labomat BFA (Mathis, Oberhasli, Switzerland), using mini reactors of 100 mL at 25 rpm. The temperature was raised to the set temperature and left for the set time, and then left to cool down to room temperature. The knitted fabric samples were 20 cm^2^.

Drying: All samples (padding and exhaustion) were dried at 100 °C for 6 min and fixed at 150 °C for 3 min. Afterward, treated samples were rinsed thoroughly in warm water (≈40 °C) to remove unfixed materials and air-dried at room temperature.

### 2.4. Design of Experiments (DoE)

The DoE methodology was used to perform an improvement of the functionalization time and temperature parameters. Therefore, a complete 2^2^ factorial design (with three replicates at the central point) comprised the time and the temperature as factors and ΔE as a response. The factors ranged between 10 and 60 min and 20 and 80 °C, for time and temperature, respectively.

State-Ease^®^ ‘Design Expert’ software (version 12; Minneapolis, MN, USA) was used to establish the design and to perform the analysis of variance (ANOVA) of the model elements.

### 2.5. UV-Visible Spectroscopy

The optical properties of the AuNPs-HAp suspensions were analyzed based on absorption spectra measured on a UV-Vis-NIR spectrophotometer (Shimadzu UV-3600i; Kyoto, Japan) with a double beam, in the range between 185 and 750 nm with a spectral resolution of 1 nm. For every dispersion, the adsorption was measured at least 3 times.

The optical properties of the textile samples were analyzed based on the total reflectance spectra measured on a UV-Vis-NIR spectrophotometer (Shimadzu UV-2600i; Kyoto, Japan) coupled with an integration sphere (Shimadzu ISR-2600Plus; Kyoto, Japan) in the range between 220 and 1400 nm, with a spectral resolution of 5 nm. The color difference (ΔE) was determined by the Shimadzu Colour Analysis complementary software - LabSolutions UV-vis version 1.12, using as a reference the respective textile treated under the same conditions, but without the AuNPs-HAp. For every sample, the reflectance was measured at least 3 times, in different spots of the sample.

### 2.6. Washing Fastness

For the washing fastness assay, standard ISO 105-C06:2010 [33] was applied. Briefly, the fabric samples were washed in a Mathis Labomat BFA using mini reactors of 100 mL at 40 °C for 60 min, with 4 g/L ECE non-phosphate detergent. The samples were dried at 100 °C for 6 min. The samples were analyzed in UV-Vis before and after the washing cycle.

For AuNPs-HAp loss deduction, we used the formula below:(1)$$\mathrm{AuNPs}-\mathrm{HAp}\;\mathrm{Loss}=100\;-\frac{S_{20}\;\times \;100}{S_{0}}$$
where:

S_20_—ΔE of the sample after 20 washing cycles.

S_0_—ΔE of the sample after no washing cycles.

### 2.7. Atomic Absorption Spectroscopy (AAS)

Textile samples were dissolved in aqua regia, with a liquor-to-material ratio of 20:1. Then the solutions were analyzed in a novaAA^®^ 350 (AJ Analytical Instrumentation, Jena, Germany) to quantify the amount of gold. The equipment utilized has a detention limit (DL) of 0.09 mg/L Au and a quantification limit (QL) of 0.24 mg/mL Au. All values under QL were not quantified.

### 2.8. Transmission Electron Microscopy (TEM) Analysis

The TEM assays were carried out at the Jožef Stefan Institute. The AuNPs-HAp, in powder form, were analyzed by TEM through a JEOL JEM-2100 microscope from Jeol Ltd. at 200 kV. A beryllium support sample with double inclination was used to evaluate the AuNPs-HAp morphology. The powder was ultrasonically dispersed in water and deposited on carbon films, supported by a 300-mesh copper grid.

### 2.9. Scanning Electron Microscope (SEM) and Energy-Dispersive X-ray Spectroscopy (EDS) Analysis

Textile samples were cut into 5 mm^2^ pieces, and half of each sample was coated with gold to give them conductivity. SEM analysis was carried out in a JEOL microscope–JSM-6010LV (Jeol, Akishima, Tokyo, Japan). Chemical analysis by EDS was carried out in an X-act, Oxford Instruments INCA (Abingdon, UK).

### 2.10. Antimicrobial Activity

Microorganisms: Three different bacteria strains were used during the antibacterial analysis: *Pseudomonas aeruginosa* (American Type Culture Collection (ATCC) 27853), *Staphylococcus epidermidis* (ATCC 35984), and *Escherichia coli* (ATCC 25922). The antiviral tests were performed using *E. coli* bacteriophage MS2 (ATCC 15597-81). To quantify MS2 plate forming units (PFU), its host *E. coli* (ATCC 15597) was used.

Culture Media: Nutrient agar (NA) and nutrient broth (NB) were used as a medium for *P. aeruginosa*; Trypticase soy broth (TSB) and solid Trypticase soy agar (TSA) were used for *E. coli* and *S. epidermidis*; finally, ATCC 271 medium was used for virus and host manipulation. All reactants were purchased from Liofilchem (Téramo, Italy).

Antimicrobial Activity of AuNPs-HAp: Semi-quantitative analysis was performed by the determination of the minimum inhibitory concentration (MIC), using the method in [34]. Briefly, 100 µL of AuNPs-HAp dispersions at different concentrations (0, 0.08, 0.1, and 0.2 mg/mL) were diluted with 100 µL of 1 × 10^7^ colony-forming units (CFU)/mL of each microorganism (1 × 10^5^ PFU/mL for MS2), and their optical density was determined by analysis in a spectrophotometer, before and after incubation at 37 °C, 120 rpm, for 24 h. Then, for the quantitative analysis, 200 µL of each AuNPs-HAp suspension with each microorganism was serially diluted with the medium NA for *P. aeruginosa*, TSA for *S. epidermidis* and *E. coli*, and ATCC 271 medium for MS2 virus (diluted 5 times, in a proportion of 1:10).
(2)$$\mathrm{Optical}\;\mathrm{density}\;={OD}_{T24}-{OD}_{T0}$$
where:

OD_T24_—Optical density after 24 h.

OD_T0_—Optical density immediately after inoculation.
(3)$$\mathrm{Log}\;\mathrm{reduction}\;=\mathrm{Log}\;\left[\mathrm{control}\right]-\mathrm{Log}\;[\mathrm{exposed}]$$
where:

Control—Inoculum microorganism concentration.

Exposed—Microorganism concentration after 1 h of exposure.

Antimicrobial Activity of Functionalized Textiles: Quantitative analysis was performed by the determination of the minimum virucidal concentration (MVC) and the minimum bactericidal concentration (MBC) [35]. The MVC was based on MBC, adapted to bacteriophages according to the reference [36]. For the quantitative analysis, the CFU or PFU was counted. Testing was performed in textile samples without and with AuNPs-HAp (0.1 and 0.3 mg/mL), after one to twenty washing cycles. The methodology used was as described in [36]. Briefly, pre-inocula of *S. aureus*, *E. coli*, *P. aeruginosa*, and *E. coli* (host) were incubated overnight at 37 °C and 120 rpm. Then, 6.25 cm^2^ textile square samples were inoculated with 50 µL containing 1 × 10^7^ CFU/mL of *S. aureus*, *E. coli*, and *P. aeruginosa* in sterile phosphate-buffered saline (PBS) or 1 × 10^7^ PFU/mL of MS2 in ATCC 271. The samples were incubated for 1 h (2 h for MS2) at room temperature (RT = 21 °C). After the incubation period, 5 mL of PBS solution (or ATCC 271 medium for MS2) were added and the samples were mixed in the vortex for at least one minute. Then, 200 µL of the washing liquid was serially diluted with PBS or ATCC 271, and each dilution was plated in a solid medium. The plates containing ATCC 271 medium were previously inoculated with the *E. coli* host. The Petri dishes were incubated for an additional 20 h at 37 °C and 90% humidity, after which the CFU or PFU were counted. The materials were classified as presented in the Table 2.

## 3. Results

### 3.1. AuNPs-HAp Characterization

UV-Vis spectroscopy (Figure 2) of the AuNPs-HAp denoted a maximum absorbance peak of 562.5 nm. A typical peak between 558 and 568 nm is obtained in UV-Vis spectra of AuNPs’ incitation of surface plasmon resonance (SPR) [38]. Thus, the presence of HAp in the AuNPs-HAp did not interfere with AuNPs’ characteristic SPR.

Figure 3 displays the captured image from TEM, where it is possible to see the HAp support and the deposited AuNPs. The AuNPs’ size is around 20 nm.

The antimicrobial activity of AuNPs-HAp per se against several microorganisms was evaluated. All microorganisms were incubated with different concentrations of AuNPs-HAp for 24 h. The bacteria’s optical density was measured (Figure 4a). The controls (dispersions without AuNPs-HAp) denote a clear growth of the tested bacteria, whereas for all concentrations of AuNPs-HAp the turbidity is nearly not perceivable.

Furthermore, all AuNPs-HAp concentrations produced a relevant log reduction against all tested microorganisms (Figure 4b).

### 3.2. Textile Functionalization

#### 3.2.1. Surfactant Impact

Different surfactants were added to the AuNPs-HAp suspensions to improve their homogeneity, and thus improve their distribution, i.e., color uniformity, in the textile samples. The surfactants tested are common surfactants used in the textile industry as described in Section 2.1. This evaluation was performed using exhaustion, at 60 °C for 60 min in half-bleached CO.

First, the reflectance spectra of the treated textile fabrics with each surfactant were analyzed in three different areas of the CO to evaluate uniformity. Surfactant A exhibited improved reproducible results, with three overlapping spectra. The spectra of surfactants B and C did not overlap, meaning that the functionalization was not uniform (graphs and Appendix A).

The ΔE indicated better results for surfactant A before washings; however, the same sample had a higher loss of AuNPs-HAp during washing cycles. This resulted, after 20 washing cycles, in similar ΔE for surfactants A, B, and C. Surfactant B exhibited less adsorption of AuNPs-HAp to the knitted CO, but higher fastness to washings (Figure 5).

Given these results, further tests were performed with the two surfactants that showed better results: A and B. Surfactants A and B were separately combined with surfactant D, to study the synergy. The results, compared to using just one surfactant, improved greatly, both in terms of color uniformity and intensity (Figure 6).

#### 3.2.2. Pre-Treatments

Several pre-treatments were tested to improve the affinity of the AuNPs-HAp to the used knitted fabrics, using the combination of surfactants A and D. The raw CO knitted fabric was not tested due to its inherent hydrophobicity, which considerably impairs functionalization using an aqueous dispersion [39]. CO knitted fabric was pre-treated with half-bleaching and/or DBD plasma. The CO–PES and PES knitted fabric were pre-treated with DBD. The half-bleach pre-treatment was not applied to CO–PES and PES knitted fabric since they were already bleached and dyed. In addition, half-bleaching is mostly used in cotton fabrics to remove impurities and confer them some whiteness to improve the dyeing process [39]. The knitted fabrics, with and without pre-treatment, were functionalized with AuNPs-HAp, and submitted to up to 10 washing cycles.

Figure 7 shows the ΔE of the knitted fabrics of all treatments; half-bleaching was the one that resulted in more AuNPs-HAp bonded to the textile, even after 10 washes. Moreover, the figure also shows that, although the CO–PES untreated fabric without washing had better adsorption of AuNPs-HAp (slightly higher ΔE than fabric treated with DBD), this sample had a much higher loss of AuNPs-HAp during washing cycles, in opposition to the fabric treated with DBD, which showed a good fastness through 10 washing cycles. For the PES fabric, both samples (untreated and treated) show some loss of AuNPs-HAp during washing cycles, but the better results (higher number of AuNPs-HAp, either before or after 10 washing cycles) were shown by the DBD-treated fabric.

#### 3.2.3. Functionalization Process

The half-bleached CO knitted fabric clearly showed a higher affinity for the AuNPs-HAp with surfactants A and D. Therefore, for further optimization, additional variables and processes were studied to optimize functionalization yields and washing fastness.


Functionalization Process: Exhaustion vs. Padding


During the comparison of the exhaustion and padding functionalization processes, exhaustion was performed at 60° C and padding was performed at room temperature. Figure 8 displays the AuNPs-HAp absorption into the fabric according to the two functionalization methods. Exhaustion showed better results, namely higher initial bonding of the AuNPs-HAp and lower loss during washing cycles (12.6% loss against 28% loss for the padding process). Thus, the exhaustion methodology was further optimized in the following steps.


Liquor-to-Substrate Ratio


To mitigate costs and increase the efficiency of the exhaustion process, different liquor-to-substrate ratios were tested: 20:1, 12:1, and 7:1. The most homogenous absorption was achieved in the 20:1 ratio (which was used in the previous exhaustion tests). A decrease in the ratio led to staining and low adsorption of AuNPs-HAp by the knitted fabrics. It was possible to notice after visual inspection that the dispersions collected after the exhaustion process with ratios 12:1 and 7:1 still had high concentrations of AuNPs-HAp, meaning that they were not completely adsorbed by the knitted fabrics. This was corroborated by the absorbance spectra (Figure 9) (Appendix A).


DoE


DoE was applied to optimize the exhaustion process, in particular its time and temperature, which are highly relevant parameters for the textile industry. This methodology allows for the construction of a model that predicts the outcomes of several time and temperature combinations with a minimal number of experiments [40]. These assays were carried out by exhaustion with 0.1 mg/mL of AuNPs-HAp and a combination of surfactants A and D. Time and temperature were varied as indicated in Table 1. 

The analyzed response was the ΔE and the obtained orthogonal model displayed the following fitness parameters: a standard deviation of 0.44, a mean of 10.95, an R^2^ of 0.98, an adjusted R^2^ of 0.96, a coefficient of variance of 4.05%, and an adequate precision of 16.04. Table 3 displays the ANOVA results of the orthogonal design.

The equation of the obtained model is shown in Equation (4).
(4)ΔE=4.123+0.074 Time + 0.111 Temperature −0.001 Time × Temperature

The temperature was the significant factor of exhaustion, with ΔE increasing as the temperature increased. During the tested period, time did not impact the exhaustion process, nevertheless, the interaction between the time and temperature is nearly significant. The model may be observed as a contour plot in Figure 10.

According to the DoE, higher temperatures will enhance AuNPs-HAp functionalization; nevertheless, to mitigate textile industry costs, the following conditions were selected: 70 °C for 10 min. The assay was repeated under these conditions to assess model validity (Figure 10).

In addition to the DoE, all the samples obtained were subjected to several washing cycles. Both ΔE and color strength of the textile sample increased after exhaustion at 70 °C for 10 min, while the total reflectance decreased, as did the absorbance of the dispersion used in the process. This sample also showed a slightly lower loss of AuNPs-HAp during washings with circa 28% of AuNPs-HAp lost after 20 washing cycles, an acceptable value for this type of functionalization (Figure 11).


Influence of Different Knitted Fabrics


After applying the AuNPs-HAp in half-bleached CO, the other knitted fabrics were functionalized: CO–PES and PES after DBD plasma pre-treatment.

As expected, the half-bleached CO was the fabric with the greatest adsorption of AuNPs-HAp, as the ΔE was higher (Figure 11). The PES fabric showed good adsorption of AuNPs-HAp but low washing fastness, as the fabric lost approximately 70% of AuNPs-HAp after 20 washes. The CO–PES knitted fabric showed some AuNPs-HAp adsorption. However, the UV-Vis analysis in this fabric was difficult, as it already had a green color (Figure 12) (Appendix A).

#### 3.2.4. Textiles Characterization

AuNPs-HAp concentration in the knitted fabrics was quantified after functionalization in optimal conditions (exhaustion at 70 °C for 10 min, with dispersions containing 0.1 of AuNPs-HAp, 4 g/L of surfactant A, and 5 g/L of surfactant D) through AAS. In addition, their antimicrobial efficacy was determined.


AAS


AAS was used to quantify the gold present in the textile knitted fabrics functionalized with 0.1 mg/mL of AuNPs-HAp after one and twenty washing cycles. Table 4 shows the gold quantity present in each sample.

The CO knitted fabrics displayed a concentration of Au nearly 1.3-fold higher in comparison to CO–PES and PES. CO functionalized knitted fabric lost, between the 1st and 20th washing cycle, 23.8% of Au. The CO–PES roughly lost 50% of the Au between the washing cycles. Moreover, PES knitted fabrics exhibited a percentage of Au loss of nearly 67%, denoting the weakest washing fastness, underscoring the low binding of the AuNPs-HAp to PES knitted fabric.


SEM and EDS Analysis


Knitted fabrics functionalized through the optimal conditions before and after one washing cycle were analyzed through SEM and EDS. EDS analysis indicated that the CO knitted fabric showed the highest concentration of AuNPs-HAp (Figure 13a). A lower concentration and distribution uniformity of the AuNPs-HAp was observed in the CO–PES and PES samples (Figure 13b,c).

SEM analysis indicated that the CO knitted fabric showed the highest amount of AuNPs-HAp (Figure 14a–c). The fibers showed a well-spread distribution of the AuNPs-HAp clusters. Magnification of the clusters displays the presence of 30 nm AuNPs. The higher dimension values measured in SEM compared to the TEM analysis in Figure 2 can be attributed to the Au coating and the apatite layer. In the CO–PES fabric, is possible to see the presence of bigger clusters and a lower distribution uniformity of the AuNPs-HAp. Higher magnification showed the presence of a few HAp clusters with lower amounts of AuNPs (Figure 14d–f). PES fabric displays the lowest amount of AuNPs-HAp (Figure 14g–i). Almost no AuNPs can be detected on the surface of the fibers.


Antimicrobial Properties Evaluation


Antimicrobial properties were evaluated in all knitted fabrics functionalized through optimized conditions before and after multiple washing cycles.

Figure 15 represents the fabrics’ antimicrobial activity towards the *E. coli* bacterium. The CO samples showed antibacterial activity, with a log reduction above 5, and thus with strong disinfectant properties, after one washing cycle, and log reductions between 2 and 3 after ten and twenty washing cycles, meaning that the samples maintained strong decontaminant properties. For CO–PES and PES samples, the antimicrobial activity remained between samples, with log reductions between 2 and 3, meaning that these samples are classified as strong decontaminants.

For *S. epidermidis*, it is possible to observe, in Figure 16, that the functionalized CO had the highest log reduction (values between 2 and 5 in all samples) meaning that this fabric goes from a moderate disinfectant to a strong decontaminant after 20 washing cycles. CO–PES and PES knitted fabrics had log reductions of between 2 and 3 during all washing cycles, meaning that these are strong decontaminants. 

The antimicrobial activity of the knitted fabrics against *P. aeruginosa* is displayed in Figure 17. For CO samples, the log reduction was around 2 after one washing cycle and there was a small decrease after ten and twenty washing cycles, meaning that the samples went from strong decontaminants to weak decontaminants. For CO–PES samples, the log reduction remained between 2 and 3 during all the washing cycles, meaning that the samples had strong decontamination properties. The PES fabrics had antimicrobial activity after one and ten washing cycles, with log reductions between 2 and 3. However, after 20 washing cycles, the antimicrobial activity decreased and the log reduction was under 1, losing its antimicrobial properties against *P. aeruginosa*.

Figure 18 shows the antiviral activity of all the samples against the bacteriophage MS2. The CO–PES and PES knitted fabrics functionalized with AuNPs-HAp had a log reduction of nearly 2, after one, ten, and twenty washing cycles; thus, the fabrics are weak decontaminants. The antiviral activity of the CO knitted fabrics functionalized with AuNPs-HAp increased between the samples with one and twenty washing cycles. With one and ten washing cycles, the log reduction stayed between 3 and 4, and after twenty washing cycles, the log reduction was nearly 6, meaning that the samples went from weak disinfectants to strong disinfectants.

## 4. Discussion

The AuNPs-HAp per se have clear antimicrobial activity, exhibiting a clear inhibition of the growth of all tested bacteria. Nevertheless, the MIC was not identified due to the complete inhibition within the tested concentration range. Furthermore, MBC was also not found for *E. coli*, *S. epidermidis,* and *P. aeruginosa* due to the higher than weak disinfectant properties of all the tested concentrations [37].

Regarding the functionalization of knitted fabrics with AuNPs-Hap, several parameters were analyzed to maximize the AuNPs-HAp’s content and improve washing fastness. Firstly, surfactants were used to enhance the dispersion of the AuNPs-HAp suspensions and improve the homogeneous absorption onto the knitted fabrics. The best performing formulation was the combination of surfactants A (4 g/L) and D (5 g/L). This combination increased the AuNPs-HAp concentration and ensured a homogeneous dispersion in the knitted fabrics. Surfactant A and D are both non-ionic, whereas B is anionic with an aliphatic ester base [41,42,43]. The negative charges provided by surfactant B may have resulted in the agglomeration of the AuNPs-HAp in the solution (images in Appendix A). Surfactant C, similarly to surfactant A, is non-ionic; however, it was designed to be used for textile printing (according to the material data sheet), which may be the reason for the lack of AuNPs-HAp homogeneity when this surfactant was used.

Raw CO is highly hydrophobic, with contact angles above 100°, and requires a pre-treatment prior to any aqueous-based functionalization or dying process [39,44,45]. Therefore, CO knitted fabrics were subjected to pre-treatments. Half-bleached CO knitted fabric displayed more than two-fold higher ΔE than CO pre-treated with DBD plasma. Nevertheless, CO DBD plasma treatment improved the washing fastness (around 16% compared to 18% in half-bleached CO). Half-bleaching promoted the cleaning of the fibers and accessibility to functional groups, allowing the adsorption of AuNPs-HAp [46]. DBD plasma is known for generating transient chemical groups at the surface of the fibers without compromising their bulk properties [47,48]. CO–PES with and without DBD plasma pre-treatment displayed similar values in ΔE graphics (between 5.5 and 4), however the loss of AuNPs-HAp after 10 washing cycles was nearly none when the knitted fabric was subjected to DBD plasma. In non-pre-treated CO–PES, the AuNPs-HAp loss during washing was nearly 60%. The DBD plasma pre-treatment on PES knitted fabrics enhanced the ΔE four-fold and improved washing fastness, proving the efficacy of the pre-treatment. DBD treatment generated functional groups, allowing a higher interaction between the AuNPs-HAp and the knitted fabric surfaces through different chemical bonds, such as covalent bonding, hydrogen bonds, van der Waals forces, and dipolar interactions [17,31], which improved the load and washing fastness.

Comparing the exhaustion and padding processes, it was possible to see that the first was better in terms of AuNPs-HAp adsorption. The variables of time and temperature can influence the results obtained between these two processes. The exhaustion process allows the use of a higher temperature, more contact time, and more mechanical actions between the AuNPs-HAp and the fabric, which can increase the adsorption of the NPs [49,50]. It was also observed, upon visual inspections, that the samples obtained by exhaustion had a more uniform color distribution than the ones obtained by padding, which can be explained by the low dispersion of the AuNPs-HAp in the suspensions used in the padding process. This effect is less pronounced during exhaustion as the suspension is subjected to constant stirring. Regarding the liquor-to-substrate ratio, three liquor ratios were tested: 20:1, 12:1, and 7:1. The samples showed that, as the liquor ratio decreased, the ΔE also decreased. The AuNPs-HAp’s concentration remained on the lower liquor ratios, meaning that there was a lower quantity of NPs adsorbed by the knitted fabrics [45]. The samples functionalized with these liquor ratios also showed some lack of uniformity. This may confirm that the water reduction can lead to the AuNPs-HAp’s aggregation and poor migration ability [45]. 

The DoE assay provided a significant model, with an R^2^ and adjusted R^2^ very close to unity. The percentage of coefficient of variation is a measure of the amount of variation as a percentage of the total mean; thus, the model is reproducible due to its value of more than two-fold less than 10%. Furthermore, adequate precision displays a high value in the certification [40]. Therefore, this model can be used to navigate the parameter of time and temperature. The temperature was the significant factor; nevertheless, it seems that the ANOVA showed that temperature is the most significant variable, in comparison with time [51], which could be validated in practice. This means that the AuNPs-HAp’s adsorption increases as the temperature increases. However, above 70 °C, there was no increase in AuNPs-HAp adsorption in the test condition. The processing time was shortened to reduce costs and save energy. Therefore, achieving maximum efficiency of the exhaustion process was possible at 70 °C for 10 min.

In general, as shown in the UV-Vis results (Figure 12), the AAS (Table 4), and the SEM results (Figure 14), the AuNPs-HAp adsorption was higher in the half-bleached CO knitted fabric, and this was also the knitted fabric with the highest washing fastness results. The half-bleached CO knitted fabric had a six-fold higher ΔE compared to the PES knitted fabric and a 12-fold higher ΔE compared to the CO–PES knitted fabric in the treated samples without any washing cycle. In addition, the half-bleached CO sample lost a lower amount of AuNPs-HAp during washes, losing around 21% compared to PES (70%) and CO–PES (72%). This may indicate that, although the initial adsorption of AuNPs-HAp had promising results, the NPs created weak bonds with the PES knitted fabric, making them easy to release from the knitted fabric during mechanical actions. Likewise, the CO–PES had a similar behavior and lost a large amount of AuNPs-HAp present on the knitted fabric during the washing cycles. It seems that these AuNPs-HAp do not bond with PES fibers. Nevertheless, it is necessary to highlight that, as said before, these knitted fabrics underwent a plasma treatment, which only modifies the surface of the fabrics. For this reason, it is expected that less AuNPs-HAp bond to these knitted fabrics, compared to the CO substrate, where bonds may occur throughout the whole knitted fabric. In addition, PES fibers do not have fibrils or intrafibrillar space. For this reason, the AuNPs-HAp adsorption on PES fibers is lower than on CO fibers.

The functionalized fabrics’ antimicrobial properties were shown in Figure 15, Figure 16, Figure 17 and Figure 18. The authors studied the antimicrobial properties of knitted fabrics functionalized with NPs without washing cycles. The relevant variation of the antimicrobial activity may be a reflection of the AuNPs-HAp modus operandi, since its antimicrobial activity may imply a direct contact with the microorganism and the AuNPs-HAp. This direct interaction may in turn inactivate that specific group of AuNPs, which will result in a more pronounced uneven activity. The treated fabrics presented log reductions between 0.50 and 2 [52] or 4 [53], for *S. aureus* and *E. coli*. Other authors studied how washing cycles affected the antimicrobial properties of functionalized knitted CO fabrics with silver chloride NPs. These fabrics showed log reductions of 5 after zero, ten, and twenty washing cycles [54]. The knitted fabrics functionalized in this study presented log reductions above 2, including after several washing cycles. CO was the knitted fabric that consistently displayed higher antimicrobial activity against all tested microorganisms. Functionalized CO knitted fabric exhibited a reduction in its antimicrobial efficacy with increasing washing cycles, except against MS2 bacteriophage, where the virucidal properties slightly increased after 20 washing cycles. In opposition, the CO–PES and PES functionalized knitted fabrics displayed lower antimicrobial activity than CO, except against *P. aeruginosa*, where similar values were observed. Interestingly, CO–PES and PES displayed slightly higher activity against *P. aeruginosa* and less loss of efficacy after 10 and 20 washing cycles. Nevertheless, functionalized CO–PES and PES did not exhibit clear activity against *S. epidermidis*.

## 5. Conclusions

Three different knitted fabrics (CO, CO–PES, and PES) were functionalized with AuNPs-HAp to obtain a comfortable medical textile for a safer sharing of these textiles, and even to promote a reduction in the environmental microbial load of hospitals and nursing homes. The methodologies optimized denote methodologies that can be easily implemented by the textile industry to achieve CO knitted fabrics with antimicrobial properties against several bacteria and a non-enveloped bacteriophage. Furthermore, the antimicrobial activity remains relevant after 10 and 20 washing cycles. To further ensure their safe application, sensitization tests and skin penetration tests will be performed; nevertheless, both Au and HAp are promising materials.

## Figures and Tables

**Figure 1 nanomaterials-13-01752-f001:**
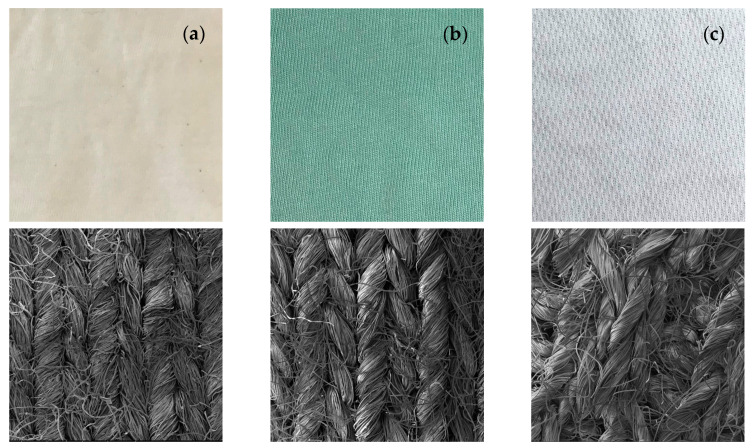
Optical and SEM images of (**a**) raw CO knitted fabric; (**b**) CO–PES knitted fabric dyed in green; and (**c**) PES knitted fabric with 100× magnification.

**Figure 2 nanomaterials-13-01752-f002:**
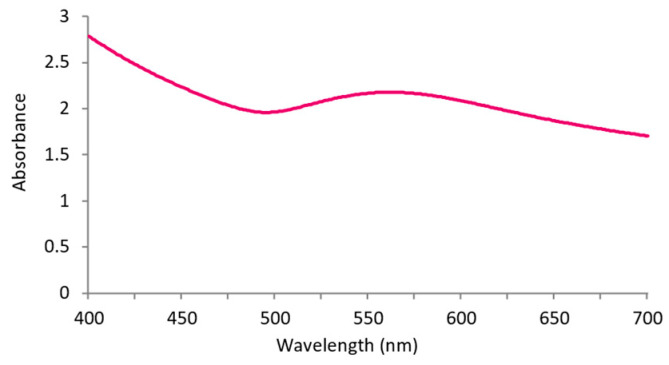
Absorbance of AuNPs-HAp (0.4 mg/mL) in distilled water (*n* = 3).

**Figure 3 nanomaterials-13-01752-f003:**
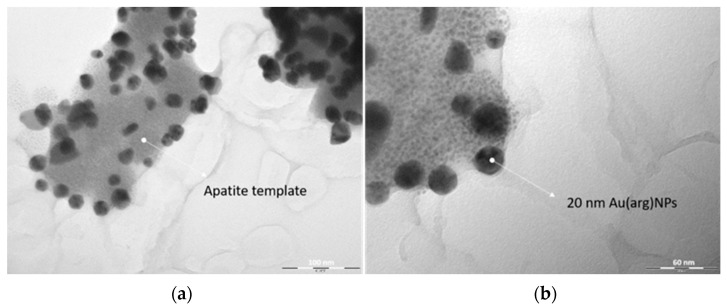
AuNPs-HAp morphology and structure through TEM. (**a**) AuNPs-HAp: the arrow indicates the HAp; (**b**) AuNPs-HAp with higher magnification, and the arrow denotes the Au(arg)NPs.

**Figure 4 nanomaterials-13-01752-f004:**
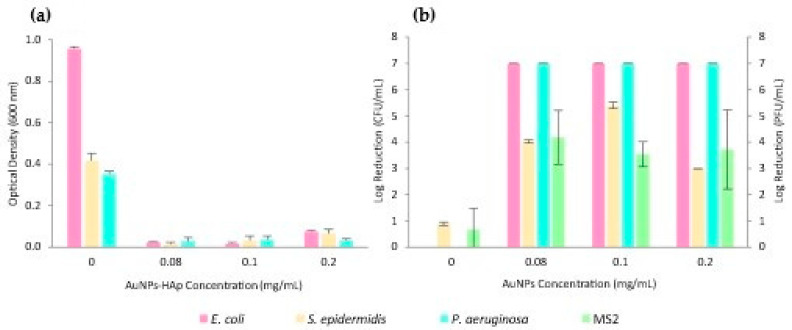
(**a**) MIC test of the AuNPs-HAp and microorganisms’ dispersions after 24 h (*E. coli*, *S. epidermidis*, and *P. aeruginosa*); (**b**) AuNPs-HAp MVC test (MS2) and MBC assay (*E. coli*, *S. epidermidis*, and *P. aeruginosa*).

**Figure 5 nanomaterials-13-01752-f005:**
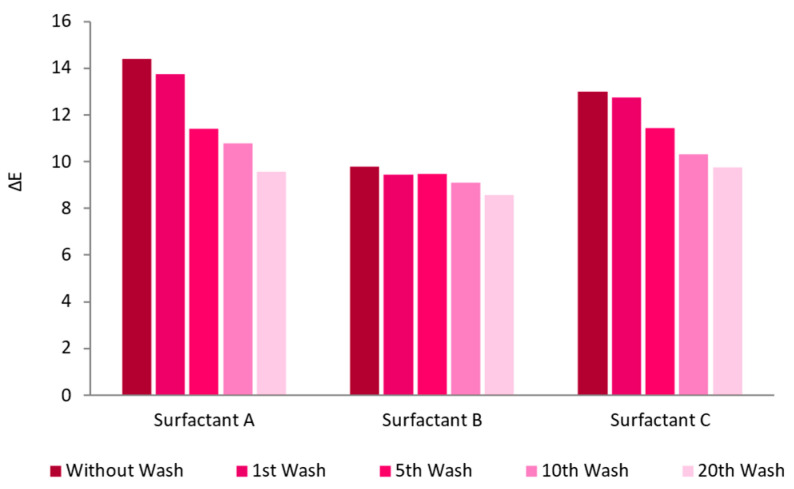
Washing cycles’ influence on the ΔE of CO knitted fabrics functionalized with AuNPs-HAp (0.1 mg/mL dispersions) with different surfactants by exhaustion (at 60 °C for 60 min).

**Figure 6 nanomaterials-13-01752-f006:**
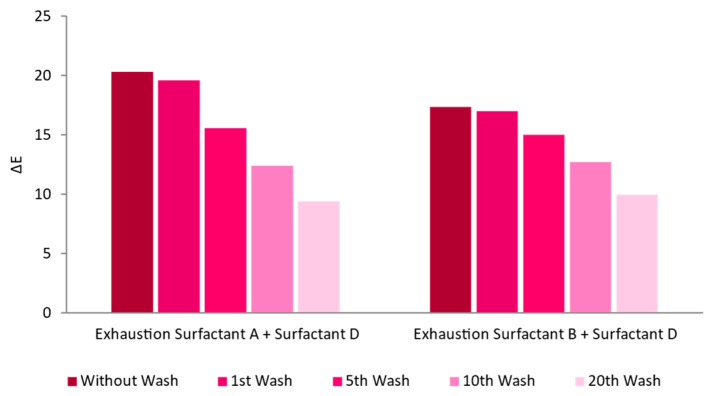
Washing cycles’ influence on the ΔE of CO knitted fabrics functionalized with AuNPs-HAp (0.1 mg/mL dispersions) with two different combinations of surfactants (at 60 °C for 60 min).

**Figure 7 nanomaterials-13-01752-f007:**
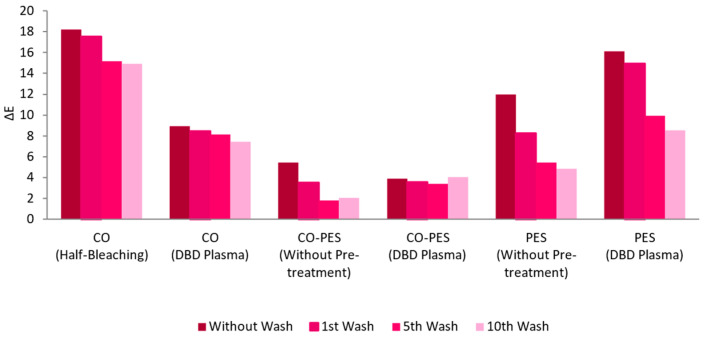
Washing cycles’ influence on the ΔE of functionalized samples (treated with 0.1 mg/mL of AuNPs-HAp dispersion, 4 g/L of surfactant A, and 5 g/L of surfactant D, by exhaustion at 70 °C for 10 min) with different pre-treatments.

**Figure 8 nanomaterials-13-01752-f008:**
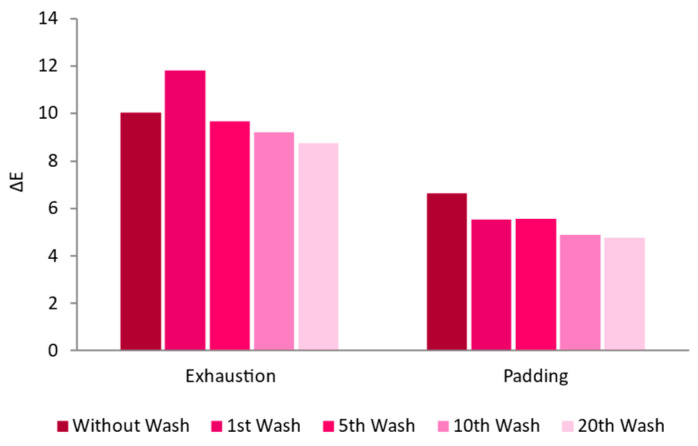
Washing cycles’ influence on ΔE for CO samples functionalized with different treatments (padding at room temperature or exhaustion at 60 °C for 60 min with 0.1 mg/mL of AuNPs-HAp).

**Figure 9 nanomaterials-13-01752-f009:**
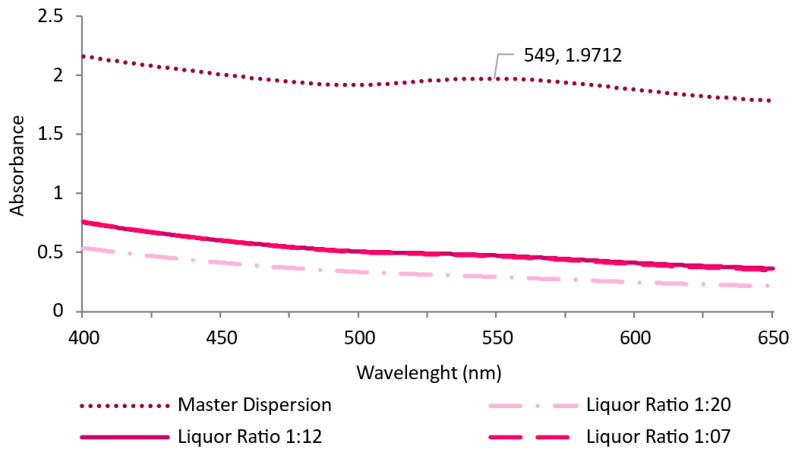
Absorbance of the dispersions after exhaustion processes (with 0.1 mg/mL of AuNPs-HAp and 4 g/L of surfactant A at 60 °C for 60 min) with different bath ratios.

**Figure 10 nanomaterials-13-01752-f010:**
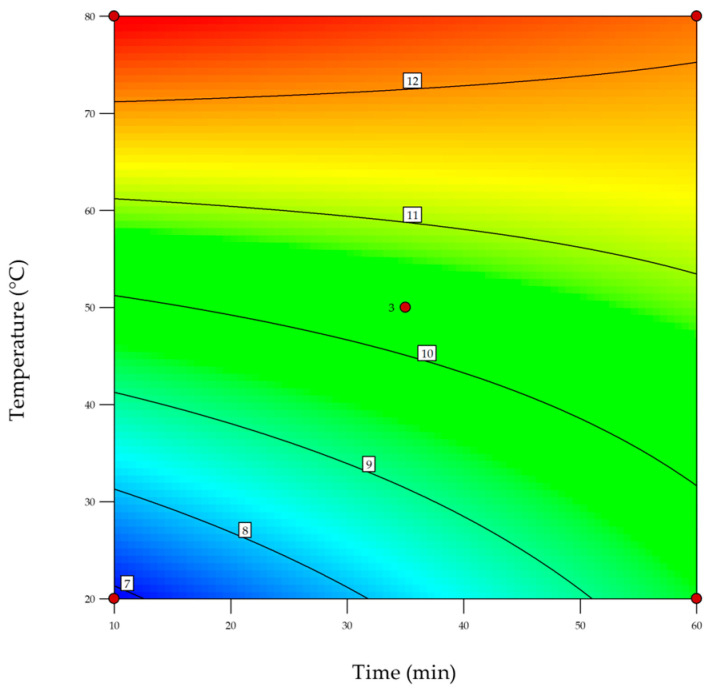
Contour plot of the effects of time and temperature during the exhaustion of AuNPs-HAp in half-bleached CO using surfactants A and D.

**Figure 11 nanomaterials-13-01752-f011:**
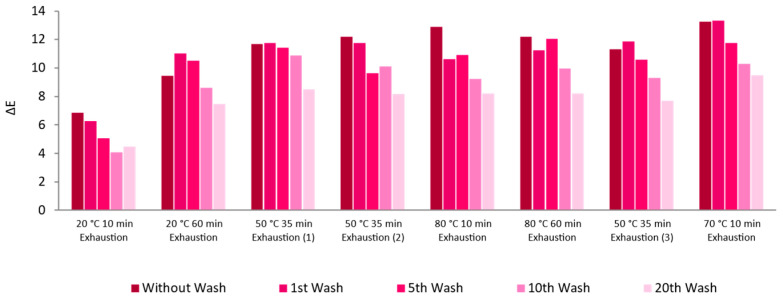
Washing cycles’ influence on the ΔE of CO knitted fabrics functionalized with AuNPs-HAp (0.1 mg/mL) and surfactant (4 g/L of surfactant A and 5 g/L of surfactant D) dispersions in different conditions (time and temperature) by an exhaustion process.

**Figure 12 nanomaterials-13-01752-f012:**
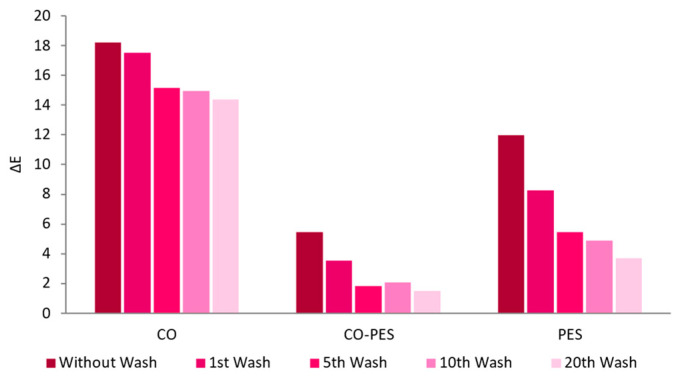
ΔE before and after washing cycles of all knitted fabrics (CO, PES, and CO–PES) functionalized (with 0.1 mg/mL of AuNPs-HAp, 4 g/L of surfactant A, and 5 g/L of surfactant D at 70 °C for 10 min) by an exhaustion process.

**Figure 13 nanomaterials-13-01752-f013:**
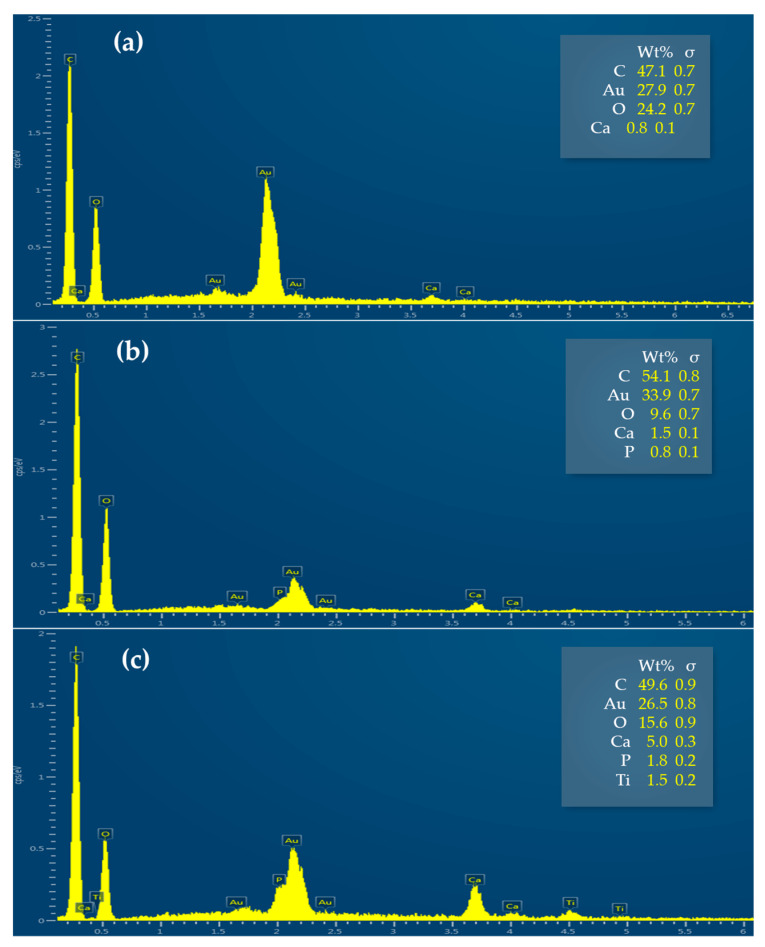
EDS spectra of the (**a**) CO, (**b**) CO–PES, and (**c**) PES knitted fabrics functionalized by exhaustion (with 0.1 mg/mL of AuNPs-HAp, 4 g/L of surfactant A, and 5 g/L of surfactant D at 70 °C for 10 min).

**Figure 14 nanomaterials-13-01752-f014:**
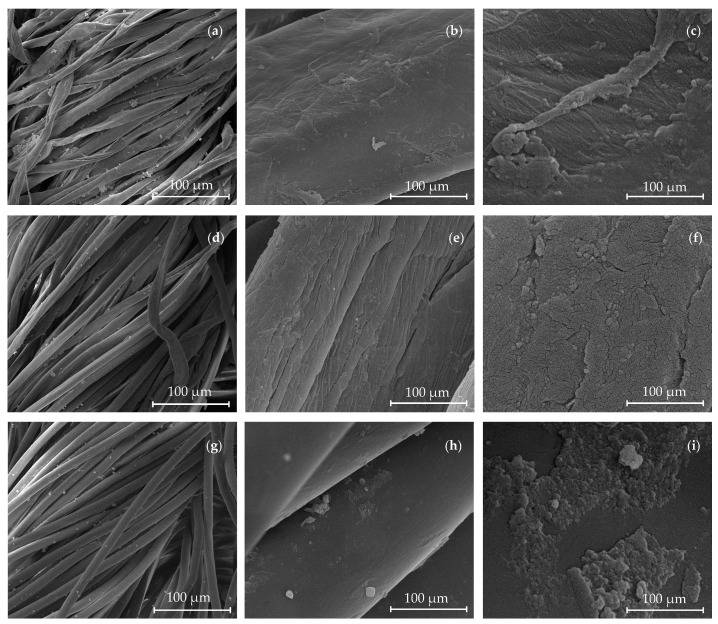
SEM images with 1000×, 15,000×, and 100,000× magnifications of the (**a**–**c**) CO, (**d**–**f**) CO–PES, and (**g**–**i**) PES knitted fabrics functionalized by exhaustion (with 0.1 mg/mL of AuNPs-HAp, 4 g/L of surfactant A, and 5 g/L of surfactant D at 70 °C for 10 min).

**Figure 15 nanomaterials-13-01752-f015:**
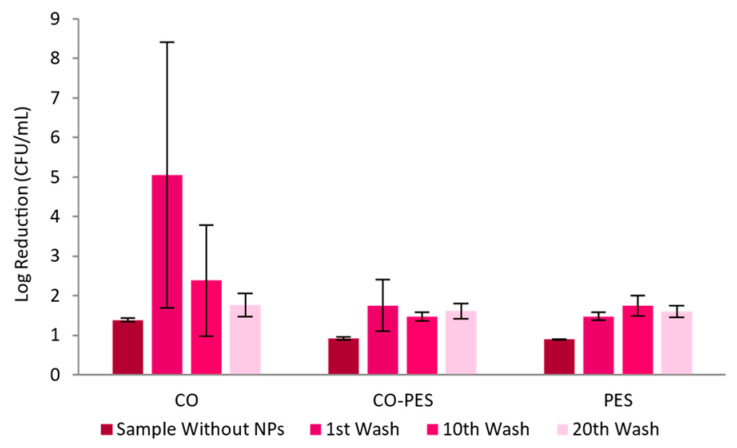
Adapted AATCC 100 assay—Log reduction in *E. coli* by different knitted fabrics functionalized by exhaustion with 0.1 mg/mL AuNPs-HAp.

**Figure 16 nanomaterials-13-01752-f016:**
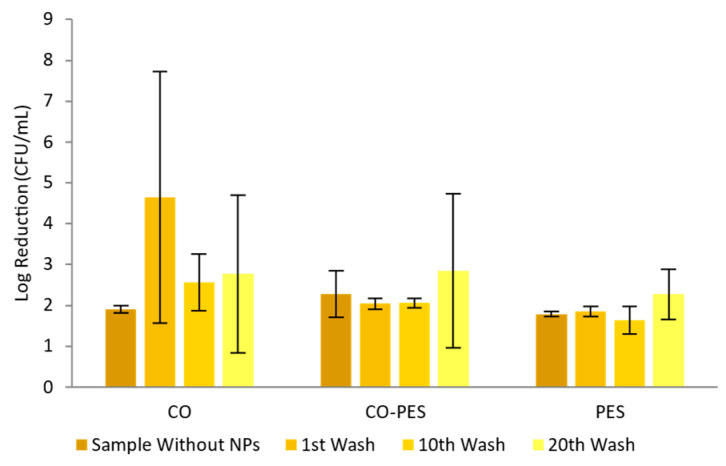
Adapted AATCC 100 assay—Log reduction of *S. epidermidis* by different knitted fabrics functionalized by exhaustion with 0.1 mg/mL of AuNPs-HAp.

**Figure 17 nanomaterials-13-01752-f017:**
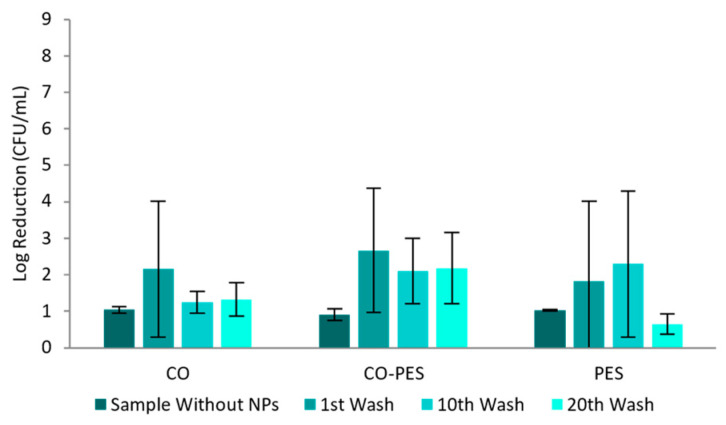
Adapted AATCC 100 assay—Log reduction in *P. aeruginosa* by different knitted fabrics functionalized by exhaustion with 0.1 mg/mL of AuNPs-HAp.

**Figure 18 nanomaterials-13-01752-f018:**
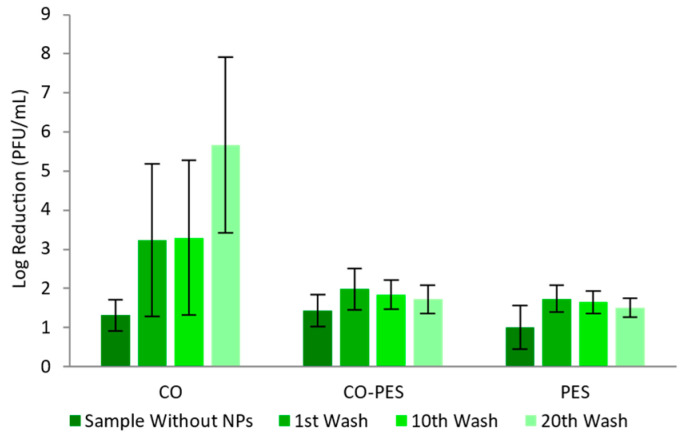
Merged and adapted AATCC 100 and ISO 18184 methods—Log reduction in different knitted fabrics functionalized by exhaustion with 0.1 mg/mL of AuNPs-HAp (with 4 g/L of surfactant A and 5 g/L of surfactant D at 70 °C for 10 min), in the presence of *E. coli* MS2 bacteriophage.

**Table 1 nanomaterials-13-01752-t001:** Process conditions and additive quantities.

Process	AuNPs-HAp Concentration (mg/mL)	pH	Temperature (°C)	Time (min)	Bath Ratio	Additives
Padding	0.1; 0.3	6; 9	Room temperature	-	-	Na_2_CO_3_ (pH adjustment). Surfactants A (4 g/L), B (0.5 g/L), C (3 g/L), and D (5 g/L)
Exhaustion	0.1; 0.3	6; 9	20–80	10–60	7:1–20:1	

**Table 2 nanomaterials-13-01752-t002:** Correlation between viability Log10 reduction and viability reduction, as well as their qualitative classifications (adapted from [37]).

Viability Log_10_ Reduction	Viability Reduction (%)	Classification
Log_10_ < 1	<90	No activity
1 ≤ Log_10_ < 2	90 ≤ R < 99	Weak decontaminant
2 ≤ Log_10_ < 3	99 ≤ R < 99.9	Strong decontaminant
3 ≤ Log_10_ < 4	99.9 ≤ R < 99.99	Weak disinfectant
4 ≤ Log_10_ < 5	99.99 ≤ R < 99.999	Moderate disinfectant
5 ≤ Log_10_ < 6	99.999 ≤ R < 99.9999	Strong disinfectant
6 ≤ Log_10_	99.9999 ≤ R	Sterilizing

**Table 3 nanomaterials-13-01752-t003:** DoE ANOVA parameters.

Source	Sum of Squares	df	Mean Square	F-Value	*p*-Value	
**Model**	22.82	3	7.61	38.62	0.0253	significant
Time	0.93	1	0.93	4.74	0.1613	
Temperature	19.21	1	19.21	97.56	0.0101	significant
Time and Temperature	2.67	1	2.67	13.55	0.0665	
Curvature	3.24	1	3.24	16.46	0.0557	
**Pure Error**	0.3939	2	0.1969			
**Cor Total**	26.45	6				

**Table 4 nanomaterials-13-01752-t004:** Au quantity (mg/L) present in samples functionalized by exhaustion (with 0.1 mg/mL of AuNPs-HAp, 4 g/L of surfactant A, and 5 g/L of surfactant D at 70 °C for 10 min).

Sample	Au after 1 Washing Cycle (mg/L)	Au after 20 Washing Cycles (mg/L)
CO	10.5	8.0
CO–PES	7.5	3.8
PES	8.5	2.8

## Data Availability

Not applicable.

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
