# Peer review of "Enhancing Functionalization of Health Care Textiles with Gold Nanoparticle-Loaded Hydroxyapatite Composites"

_nanomaterials, 2023, doi:10.3390/nano13111752_

Round 1

Reviewer 1 Report (Previous Reviewer 1)

2. AuNPs-HAp was provided and there is no originality of the authors in the material which is the origin of the functions.

R2: The AuNPs-HAp were kindly provided by Jožef Stefan Institute and are patented. These are novel materials, provided to our group in exclusivity, and their adequate incorporation into knitted textiles is the key novelty of this manuscript. The composite contains gold nanoparticles functionalized with amine and carboxyl groups and they are immobilized in nanosized hydroxyapatite.

Reviewer’s comment

The authors' arguments against my comments miss the point. I questioned the originality of the material, but the authors did not argue. The authors responded that the originality of this material lies with a company. In addition, please note that patents are for commercial purposes. It is completely pointless to use patents as a basis for scientific novelty.

3. The role of Au in this application is understandable. However, the role of HAp is unclear. R3: The functionalized gold nanoparticles are immobilized in nano-size hydroxyapatite. The role of hydroxyapatite is to promote a homogeneous dispersion preventing AuNPs aggregation, thus providing a uniform functionalization of the textiles. Furthermore, the use of hydroxyapatite is an adequate immobilization substrate as it displays enhanced biocompatibility. This information was added to the manuscript.

Reviewer’s comment

The author mentioned that “This information was added to the manuscript.”. However, it is unclear where this was added. I have never read such an unfriendly response letter before!

4. Significant difference tests should be performed on the results of antibacterial test. R4: The antimicrobial data does not possess all the required prerequisites for parametric statistical analysis since the distributions do not have the same variance and do not display a Gaussian distribution. Thus, we decided not to perform non-parametric statistical analysis.

Reviewer’s comment

The authors' arguments are understandable. However, in general, significance tests are mandatory for experiments with cells and bacteria. It is strongly recommended to perform statistical analyses assuming a Gaussian distribution.

Author Response

Reviewer 1

  1. The authors' arguments against my comments miss the point. I questioned the originality of the material, but the authors did not argue. The authors responded that the originality of this material lies with a company. In addition, please note that patents are for commercial purposes. It is completely pointless to use patents as a basis for scientific novelty.

R1: We think that we understand the Reviewer's point of view: the Reviewer is doubting of the originality and more importantly the novelty, of a patented novel nanocomposite by an Institute. Correct? If so we partially agree with the Reviewer, but please allow us to rebut: even though the nanocomposite was not synthesized by us, the methodology followed for the functionalization and all the analysis performed to the functionalized knitted are, in our point of view, extremely relevant for the entire scientific and general community. In other words, the functionalization, the analysis results, and the future perspectives described in this manuscript have the potential to enlighten and generate novel knowledge and applications with a similar approach, using similar (or even dissimilar) nanocomposites. Therefore, this argument may be used to refute the Reviewer's statement: “It is completely pointless to use patents as a basis for scientific novelty”, our objective is to display the best-performing methodologies to functionalize knitted fabrics with this type of nanocomposite and evaluate several parameters that include its washing fastness and activity (in this case antimicrobial). Does the Reviewer understand our point of view?

  1. The author mentioned that “This information was added to the manuscript.”. However, it is unclear where this was added. I have never read such an unfriendly response letter before!

R2: We beg your pardon, we did not, in any way, intend to be unprofessional or unfriendly. Please allow us to complete the information in this reply. The information was previously added to the manuscript in Lines 74 to 78: “The role of hydroxyapatite is to promote a homogeneous dispersion preventing AuNPs aggregation, thus providing a uniform functionalization of the textiles. Furthermore, the use of hydroxyapatite composite is an adequate immobilization substrate for AuNPs as it displays enhanced biocompatibility”. With this information, we would intend to clearly state the function of the Hap role.

  1. The authors' arguments are understandable. However, in general, significance tests are mandatory for experiments with cells and bacteria. It is strongly recommended to perform statistical analyses assuming a Gaussian distribution.

R3: We would like to respectfully disagree with the Reviewer, we do not consider the parametric statistical analysis mandatory for this type of result, as observed in a wide range of publications, including in this specific journal. Of course, we would like to perform parametric analysis, however, that was impossible, since we cannot assume a Gaussian distribution without having a Gaussian distribution, otherwise, it would be incorrect.

Reviewer 2 Report (New Reviewer)

The study presented in this manuscript is strongly-based, thoroughly performed and well presented. The applied methodology is correct. The results are well presented and explained However, several issues should be clarified before publication.

The results presented in sections 3.1 to 3.2.2. are clear. It seems that each type of  measurements were done on only one sample for each type.

Minor observation: Figs 2 and 9: there are needed more ticks on both axes. The reader should read the values, not to guess an approximate value of the wavelength and absorbance. 

Section 3.2.3 - EDS measurement were supposed to be done on different zones of each fabric. If this is true, this detail should be written. If not,  new  measurements will be necessary and the errors should be presented. Anyway, Figs. 13 should have the text inside written with larger characters.

Section 3.2.3 - Antimicrobial Properties Evaluation

The error bars seem to be just positive, which is weird. The authors should explain why or to rectify the figures, such as also the negative part of the error to be visible.

Figs. 15 - 18 Some errors bars are so large that the result is not to be trusted. 

The authors should present the number and dimension of the samples used in these experiments. Also they must comment on the errors values obtained for different samples. If the errors are too large and there is no rational explanation for this, the experiment should be repeated such as to reduce the error values.

Author Response

Reviewer 2

The study presented in this manuscript is strongly based, thoroughly performed, and well presented. The applied methodology is correct. The results are well presented and explained. However, several issues should be clarified before publication.

  1. The results presented in sections 3.1 to 3.2.2. are clear. It seems that each type of measurements were done on only one sample for each type.

R1: For UV-Vis analysis, every sample was analyzed 3 times in different spots to calculate the average color difference (ΔE). For absorbance analysis, 3 measurements were conducted for every dispersion.  Information added (lines 150 and 156-158).

Line 150: “For every dispersion the adsorption was measured at least 3 times.”

Line 156 – 158: “For every sample, the reflectance was measured at least 3 times, in different spots of the sample.”

In addition, in the figure caption, the following text was added in line 246: (n=3)

  1. Minor observation: Figs 2 and 9: there are needed more ticks on both axes. The reader should read the values, not to guess an approximate value of the wavelength and absorbance.

R2: We agree with the Reviewer's suggestion, additional ticks were added on the X and Y-axes of Figure 2 and X-axis of figure 9.

  1. Section 3.2.3 - EDS measurement were supposed to be done on different zones of each fabric. If this is true, this detail should be written. If not, new measurements will be necessary and the errors should be presented. Anyway, Figs. 13 should have the text inside written with larger characters.

R3:  We agree with the Reviewer, thus EDS was performed in deferent zones of the fabric, however the SEM-EDS software did not allow an adequate output of the results, thus a sole representative result was recorded and displayed. Furthermore, the SEM-EDS equipment is overbooked (the other SEM from our Institution is not operational), we can only perform new measurements is nearly two months. Since it represents a representative result, we would like to kindly as if it is absolutely mandatory? Furthermore, we revised the insets of Figure 13.

  1. Section 3.2.3 - Antimicrobial Properties Evaluation

The error bars seem to be just positive, which is weird. The authors should explain why or to rectify the figures, such as also the negative part of the error to be visible.

R4: We did not add the negative error bar since we consider implicit the negative part. We, and several other authors, always published our column results in this way. Nevertheless, we inserted the negative part.

  1. Figs. 15 - 18 Some errors bars are so large that the result is not to be trusted.

R5: These results may actually be a reflection of the modus operandi of the AuNPs-HAp, since its antimicrobial activity may imply direct contact with the microorganism and the AuNPs-HAp. This direct interaction may in turn inactivate that specific group of AuNPs, which will result in a more pronounced uneven activity. Thus, this variation may represent in our opinion a more accurate picture of the actual antimicrobial activity. This information was added in the manuscript in line 582 - 586: “The relevant variation of the antimicrobial activity may reflect the AuNPs-HAp modus operandi, since its antimicrobial activity may imply a direct contact with the microorganism and the AuNPs-HAp. This direct interaction may in turn inactivate that specific group of AuNPs, which will result in a more pronounced uneven activity.”

  1. The authors should present the number and dimension of the samples used in these experiments. Also, they must comment on the error values obtained for different samples. If the errors are too large and there is no rational explanation for this, the experiment should be repeated such as to reduce the error values.

R6: The dimension of the samples was 20 cm2, therefore, we consider this size to be adequate for all the performed analyses. Line 132: “The knitted fabrics had 20 cm2

Reviewer 3 Report (New Reviewer)

The authors have functionalized fabrics with novel AuNPs-Hap for antimicrobial application. The authors have optimized the antimicrobial activity of the knitted fabrics based on two pre-treatments, four different surfactants, and two incorporation processes. The authors have demonstrated the knitted CO displayed antibacterial properties even after 20 washing cycles. Overall, this work can inspire more material design ideas of functionalized nanomaterials with fabrics for antibacterial application. Therefore, I would like to recommend this work to publish in Nanomaterials. Below are some comments for the authors.

1. The authors have indicated that “AuNPs-HAp denoted a maximum absorbance 240 peak of 562.5 nm.” The optical property of surface plasmon resonance of AuNPs should be described in the main text.

2. For Figure 3, there are two TEM images. The authors should separately describe as Figure 3a and Figure 3b in the main text and also indicate in the caption of Figure 3. What is the Au(arg)NPs Figure 3. The authors also need to explain.

3. For the results of antimicrobial activity of AuNPs-Hap as shown in Figure 4a, with the increase of concentration, the antimicrobial activity is a little decrease. Can the authors explain the result?

4. For the introduction “Typically, metallic microparticles and NPs, such as silver, copper, gold, zinc, and titanium NPs have been applied in textiles”, more references could be cited to broaden the introduction.

https://doi.org/10.2147/IJN.S328767

Author Response

Reviewer 3

The authors have functionalized fabrics with novel AuNPs-Hap for antimicrobial application. The authors have optimized the antimicrobial activity of the knitted fabrics based on two pre-treatments, four different surfactants, and two incorporation processes. The authors have demonstrated the knitted CO displayed antibacterial properties even after 20 washing cycles. Overall, this work can inspire more material design ideas of functionalized nanomaterials with fabrics for antibacterial application. Therefore, I would like to recommend this work to publish in Nanomaterials. Below are some comments for the authors.

  1. The authors have indicated that “AuNPs-HAp denoted a maximum absorbance 240 peak of 562.5 nm.” The optical property of surface plasmon resonance of AuNPs should be described in the main text.

R1: We agree with the Reviewer, the following information was added in the manuscript text line 244 – 246: “A typical peak between 558 and 568 nm is obtained in UV-Vis spectra of AuNPs incitation of Surface Plasmon Resonance (SPR) [REF]. Thus the presence of HAp in the AuNPs-HAp did not interfered with AuNPs characteristic SPR.”

[REF] - 10.1016/j.enmm.2019.100270

  1. For Figure 3, there are two TEM images. The authors should separately describe as Figure 3a and Figure 3b in the main text and also indicate in the caption of Figure 3. What is the Au(arg)NPs Figure 3. The authors also need to explain.

R2: We thank the Reviewer for the comment. The caption was revised to include the adequate description: Line 253 - 254: “a) AuNPs-HAp, the arrow indicates the HAp, b) AuNPs-HAp with higher magnification and the arrow denotes the Au(arg)NPs.”

The the Au(arg)NPs of Figure 3 are explained in line 95-97: “The AuNPs-HAp were kindly provided by the Jožef Stefan Institute (Ljubljana, Slovenia) and produced as described in [26], with arginine (arg) as the organic molecule, and functionalized with amine and carboxyl groups.”

  1. For the results of antimicrobial activity of AuNPs-Hap as shown in Figure 4a, with the increase of concentration, the antimicrobial activity is a little decrease. Can the authors explain the result?

R3: We tried to explain in line 500 – 504 the Reviewers legitimate doubt: “The AuNPs-HAp per se have clear antimicrobial activity, exhibiting a clear inhibition of the growth of all tested bacteria. Nevertheless, the MIC was not identified due to the complete inhibition within the tested concentration range. Furthermore, MBC was also not found for E. coli, S. epidermidis and P. aeruginosa due to the higher than weak disinfectant properties of all the tested concentrations [36].”

  1. For the introduction “Typically, metallic microparticles and NPs, such as silver, copper, gold, zinc, and titanium NPs have been applied in textiles”, more references could be cited to broaden the introduction. https://doi.org/10.2147/IJN.S328767

R4: We thank the reviewer for this suggestion. We added the reference, as suggested, on other affirmations. We think that this reference is more suitable for the affirmation “These NPs have robust antimicrobial activity, even at a low concentration. The silver ions can compromise cell wall integrity and generate reactive oxygen species (ROS) that effectively compromise genomic and proteomic activity, promoting cell death”, as there is no mention of the nanoparticles application in textiles, but the antimicrobial mechanism of the nanoparticles, including, silver ones, are extremely well explained.

Round 2

Reviewer 1 Report (Previous Reviewer 1)

The paper has been thoroughly revised.

Author Response

Reviewer: The paper has been thoroughly revised.

R: Thank you for the revision and for accepting the paper

Reviewer 2 Report (New Reviewer)

Figs. 15 -18.

The confidence intervals for the measurements are asymmetric, but usually these are symmetric. In the type of measurements carried out in the presented study, other studies show symmetric confidence intervals (i.e. measurement value = m +/-error, meaning that the positive error and the negative error values are the same). Please explain why you present different values for the positive and negative errors, meaning you have asymmetric confidence intervals. (Understanding Confidence Intervals | Easy Examples & Formulas (scribbr.com))

Concluding, the errors are so large that the measurements are not to be trusted.

Author Response

Reviewer 2: C1: Figs. 15 -18. The confidence intervals for the measurements are asymmetric, but usually these are symmetric. In the type of measurements carried out in the presented study, other studies show symmetric confidence intervals (i.e. measurement value = m +/-error, meaning that the positive error and the negative error values are the same). Please explain why you present different values for the positive and negative errors, meaning you have asymmetric confidence intervals. (Understanding Confidence Intervals | Easy Examples & Formulas (scribbr.com)). Concluding, the errors are so large that the measurements are not to be trusted.

R1: We would like to acknowledge the Reviewer’s careful review. We do not understand what occurred, we added the negative error value as requested by the Reviewer, and we did not check the actual error that is clearly bugged. We do not have any reasonable explanation for these values, thus we just adequately inserted them in the graphs. As we previously answered in R5 of the previous round of reviews: “These results may actually be a reflection of the modus operandi of the AuNPs-HAp, since its antimicrobial activity may imply direct contact with the microorganism and the AuNPs-HAp. This direct interaction may in turn inactivate that specific group of AuNPs, which will result in a more pronounced uneven activity. Thus, this variation may represent in our opinion a more accurate picture of the actual antimicrobial activity. This information was added in the manuscript in line 582 - 586: “The relevant variation of the antimicrobial activity may reflect the AuNPs-HAp modus operandi, since its antimicrobial activity may imply a direct contact with the microorganism and the AuNPs-HAp. This direct interaction may in turn inactivate that specific group of AuNPs, which will result in a more pronounced uneven activity.””

Reviewer 3 Report (New Reviewer)

The authors have addressed all comments raised by the reviewers. Therefore, I would like to recommend this manuscript to publish as its current form in Nanomaterials.

Author Response

Reviewer: The authors have addressed all comments raised by the reviewers. Therefore, I would like to recommend this manuscript to publish as its current form in Nanomaterials.

R: Thank you for the revision and for accepting the paper

This manuscript is a resubmission of an earlier submission. The following is a list of the peer review reports and author responses from that submission.

Round 1

Reviewer 1 Report

The authors report in this paper the properties of health care textiles with AuNPs-HAp complexed. The reviewer understood the significance of this study to the health care material field. However, the reviewer is wondering the suitability of this paper for this journal in the following respects:

1. This paper does not include results of analysis of materials at the nano-level. For example, the particle size distribution data of AuNPs-HAp, and results of SEM and TEM observations at high magnification are required.

2. AuNPs-HAp was provided and there is no originality of the authors in the material which is the origin of the functions.

3. The role of Au in this application is understandable. However, the role of HAp is unclear.

4. Significant difference tests should be performed on the results of antibacterial test.

5. On page 12, the authors mentioned that “A lower concentration and distribution uniformity of the AuNPs-HAp was observed…”. However, the reviewer considers that the SEM magnification was too low to confirm distribution uniformity of the AuNPs-HAp.

Author Response

  1. This paper does not include results of analysis of materials at the nano-level. For example, the particle size distribution data of AuNPs-HAp, and results of SEM and TEM observations at high magnification are required.

R1:  The authors agree with the Reviewer's statements and added new high-resolution SEM figures. Furthermore, TEM observations were also included in the paper, for a thorough AuNPs-HAp characterization. Size distribution was obtained by TEM observation since a DLS analysis is impossible to perform in the AuNPs-Hap composite.

  1. AuNPs-HAp was provided and there is no originality of the authors in the material which is the origin of the functions.

R2: The AuNPs-HAp were kindly provided by Jožef Stefan Institute and are patented. These are novel materials, and their adequate incorporation into knitted textiles is the key novelty of this manuscript. The composite contains gold nanoparticles functionalized with amine and carboxyl groups and they are immobilized in nanosized hydroxyapatite.

  1. The role of Au in this application is understandable. However, the role of HAp is unclear.

R3: The functionalized gold nanoparticles are immobilized in nano-size hydroxyapatite. The role of hydroxyapatite is to promote a homogeneous dispersion preventing AuNPs aggregation, thus providing a uniform functionalization of the textiles. Furthermore, the use of hydroxyapatite is an adequate immobilization substrate as it displays excellent biocompatibility. This information was added to the manuscript.

  1. Significant difference tests should be performed on the results of antibacterial test.

R4: The antimicrobial data does not possess all the required prerequisites for parametric statistical analysis, in particular, the distributions do not have the same variance and do not display a Gaussian distribution. Thus, we decided not to perform non-parametric statistical analysis.

  1. On page 12, the authors mentioned that “A lower concentration and distribution uniformity of the AuNPs-HAp was observed…”. However, the reviewer considers that the SEM magnification was too low to confirm distribution uniformity of the AuNPs-HAp.

R5: In our opinion, as we increase the magnification, we limit the overview, thus it is harder to have an overview of the distribution. Nevertheless, we added higher-resolution SEM images. Moreover, a a direct proof of Au distribution is given by the EDX results that show the enhanced Au distribution on the fabric.
